# Innovative RNAi Strategies and Tactics to Tackle Plum Pox Virus (PPV) Genome in *Prunus domestica*-Plum

**DOI:** 10.3390/plants8120565

**Published:** 2019-12-02

**Authors:** Michel Ravelonandro, Ralph Scorza, Pascal Briard

**Affiliations:** 1Unite Mixte de Recherches 1332, INRA-Bordeaux, 33882 Villenave d’Ornon, CS 20032, France; Pascal.briard@inra.fr; 2Appalachian research Station, USDA-ARS; Kearneysville, WV 25443, USA; ralphscorza@gmail.com

**Keywords:** Prunus domestica, RNAi, silencing, resistance, recovery, plum pox virus stability

## Abstract

We developed an innovative RNAi concept based on two gene constructs built from the capsid gene (CP) cistron of the *Plum pox virus* (PPV) genome. First, designated as amiCPRNA, a potential molecule interfering with PPV genome translation and the second one is the ami-siCPRNA to target viral genome translation and PPV RNA replication. Following the previous engineering of these constructs in an experimental herbaceous host, they were introduced into *Prunus domestica* (plum tree) genome. Previously propagated onto a susceptible rootstock, these clones were graft-inoculated with PPV. After four dormancy cycles, and consistent with our experience of PPV infection, some clones showed a common phenomenon of silencing that can differ between the detailed plant phenotypes. Three different phenotypes were developed by the amisiCPRNA clones. First, the high resistance character shown by the amisiCPRNA plum-7 that was similar to the resistance expressed by HoneySweet plum. Secondly, a recovery reaction was developed by the two other amisiCPRNA plum-3 and plum-4 that differed from the rest, characterized as susceptible clones, among these were the amiCPRNA plums. Having assessed the behavior of these plums versus the herbaceous host accumulating the similar form of RNAi: ami-, si-, and ami-siRNA, challenging assays in perennials consistently reflect the natural context of viral genome targeting.

## 1. Introduction

PPV causes the severe disease known as sharka [1]. While control of local disease spread has been in many cases achieved through the harmonization of control measures, long distance virus spread has continued through human transport of propagative material in the last three decades [2]. Throughout the European Union countries, researchers are seeking solutions to control the disease incidence but the economic imbalance between each and other state led to the lack of harmonization. Presently, no resistant plum cultivar is being extensively used by fruit-tree growers. While the cv. Jojo plum released by German researchers is an achievement [3], the phenotypes shown by this plum tree in field release raised a scientific debate about resistance and potential virus reservoirs [4].

Many publications have shown the use of transgenic plants to produce crops resistant to virus infection [5]. As a potyvirus member, PPV provides a model responsible for a high economic disease impact [1], like *Papaya ringspot virus*, PRSV, which was successfully controlled with a biotechnological approach [5]. Significant progress was achieved by the production of a transgenic resistant plum, designated as HoneySweet [6,7,8]. In this case tremendous progress in controlling PPV via silencing was achieved [9]. RNAi also could be delivered in some systems by spraying dsRNA [10]. Over the last decade of investigation, silencing is the most effective molecular mechanism studied to control either plant viruses or insects [9,10,11,12,13].

Among the recent example, there was the new design of artificial miRNA (amiRNA) in *Arabidopsis* plants [11,14] which provided resistance to both *Turnip yellow mosaic virus* (TYMV) and *Turnip mosaic virus* (TuMV) [15]. The role of amiRNA accumulation in plants that triggers the targeted RNA template was also clarified [16,17]. It is dependent on the RNaseIII enzyme Dicer that processes the longer double-stranded RNA to yield products of 21–24 nt long. These require no further processing from Dicer and are primed to be incorporated directly into the RNA-induced silencing complex (RISC) that target virus genome recognition and cleavage. Since silencing was obviously helpful to understand the molecular processes of the plant defense system, studies about the factors involved in the mechanism appeared. Convincing demonstrations supported the correlation between the larger RNA template folding and processing by different special RNases into small RNAi. Their accumulation in plants validated the resistance phenotype conferred by RNAi [17,18,19].

Once we demonstrated that the accumulation of siRNA in *Nicotiana benthamiana* provided resistance to PPV, extensive studies in perennials were pursued [20]. We developed the concept concerning the production of amiRNAs [21] from the same precursor and the knowledge about the exploitation of PPV genome sequence through bioinformatics [22]. Under these studies, computational target predictions were selectively performed on the CP cistron [23]. Our focus was to use the CP cistron to target the PPV genome in order to control this virus in a woody perennial species [1,2]. Two gene constructs were engineered, the first designed as amiCPRNA and the second named as amisiCPRNA (artificial mi and si-RNA) of the CP cistron. As we proceeded in the past, they were previously verified in *N. benthamiana* plants [19,20] prior to the introduction in the *Prunus domestica* genome [24].

In the present study, we investigated the potential effects of these two gene constructs, amiCPRNA and amisiCPRNA to produce RNAi, and to assess their effects on PPV infection. The thinking behind these transgenic plums was to assess their effects on PPV infection in perennial plants. Studying much of these plant-virus interactions, since the discovery of the HoneySweet woody plant [6,7,8], substantively merited more attention. While little was known about the cooperative function of ami- and si-RNAs in perennials [25], we drew up an experimental strategy which brought a good understanding of the silencing in plum trees. From these studies we found that no apparently protected plants were recorded in clones only accumulating amiRNA. Conversely three clones with different phenotypes were characterized with the amisiCPRNA plums. Through these studies we were both able to discern the weakness of the amiRNA approach that needs to be improved and the robustness of silencing through the co-accumulation of both RNAi (ami and siRNA) to inhibit PPV replication in perennials.

## 2. Results

### 2.1. Transformed Plums and Molecular Characterization

After ensuring design for the amiCPRNA constructs (Figure 1), these were cloned into the pHellsgate plant transformation vector (Figure 2A). Hypocotyl slices of *P. domestica* were used for generating transgenic clones. A total of 25 plants were obtained including 10 amisiCPRNA plums and 15 amiCPRNA clones.

In further analysis of RNAi accumulated in each clone, a variable amount of RNAi was detected in amisiCPRNA-plums (Figure 2B) and expectedly, all ami-RNA plums produced amiRNA. Figure 2C only shows a few clones that have had an early development. Trusting with the results of these analyses, we were surprised that the amisiCPRNA-plum10 did not accumulate RNAi. However, the remaining clones accumulated RNAi.

### 2.2. Resistance Studies

#### 2.2.1. Behavior of Different Plum Clones

These assays were conducted through parallel studies with the resistant HoneySweet plum and the conventional BO70146 plum as controls. Using HoneySweet plum characterized by its ability to constantly resist PPV infection as an experimental control, we eliminated any difficulties in identifying resistant clones. In parallel, the use of PPV susceptible rootstocks, natural hosts of PPV, clearly provided infected controls and prevented any confusion in evaluating infection. Plant replicates varied from 3 to 10 copies (Table 2) and plant reaction to inoculation with PPV was evaluated after cold dormancy.

The fraction represented the number of plants infected. The denominator indicates the number of rootstocks diseased. Serological assays were recorded as positive with an OD value of 0.1 and greater. Below 0.1 OD value reflects the absorbance of background and the sap of the non-inoculated control plants.

The other clones that were not represented here had a low number of infected plants (only two) at the beginning. Consequently, their studies were delayed, while we produced the acceptable copy number (> three copies). Data on these is not shown since the few plants that were available for evaluation behaved like either the amiCPRNA-plum-2, -6, -8, -9, -11, -12 and -15 or the amisiCPRNA-plum-2, -6, and -10 as shown in Table 1.

#### 2.2.2. Phenotypes of the AmisiCPRNA Plums

Throughout these assays, three different phenotypes were characterized:

High resistance character of the amisiCPRNA plum-7.

Beyond the long last duration of these assays under greenhouse conditions, the ami-siCPRNA plum-7 diverged from all others. Following to the bud-breaking period wherever scions grew and increased in size, this clone appeared as singularly symptomless. Interestingly, while leaves from rootstock emerged and showed symptoms, shoots belonging to the clone were symptomless. From four repetitive dormancy cycles, the eight PPV detection assays confirmed that no transgenic scions were infected (Figure 3).

Recovery reaction of the amisiCPRNA plum-3 and -4.

When plants were removed from cold, a variable number of both clones showed sporadic symptoms. Because bud-breaking remodels plants through the development of new growth, PPV replication could sporadically appear in a few leaves (not shown). These were confirmed through serological and molecular detection (Figure 3). While the relative lower amount of PPV was detected, the virus disappeared after 2 weeks of growth. Surprisingly, from four repetitions of these cycles (shoot growth and vernalization), these few plants initially diseased developed the same recovery reaction.

Susceptibility of the amisiCPRNA plums.

In contrast to the three clones described above, the remaining clones including the amisiCPRNA plum-1,-2,-5,-6,-8,-9 and -10 that basically harbored the same amisiCPRNA construct, were diseased like the conventional, PPV susceptible clone BO70146. Within the repetitive dormancy cycles for virus resistance assays, PPV systemically moved in these susceptible plants. A few of these susceptible clones produced typical PPV symptoms of colorful mosaic, stunted leaves (not shown).

#### 2.2.3. Behavior of the amiCPRNA Plums

The assays were successfully recorded with multiple copies of each clone. In contrast to the susceptible amisiCPRNA plums with stunted growth, diseased amiCPRNA plants were not stunted but showed severe mosaic symptoms (not shown). Thought to be highly susceptible, through the OneStep RT/PCR analysis, a few clones partially recovered from the third dormancy cycle through the lesser replication of PPV in tip versus the bottom section that was fully diseased (Figure 4 and Figure 5, Table 3). These scenarios are unusual, we will discuss further whether it is a poor targeting of PPV genome or other factors.

### 2.3. Down-Regulation of the PPV Genome Replication by RNAi Silencing

Serological detection of PPV allowed to point out that a few copies of these clones, amisiCPRNAplum-3 and -4 were infected with PPV (Table 2 and Figure 3). The repetition of these analyses 15 days later allowed to confirm the recovery reaction developed by the amisiCPRNAplum-3, -4 and the high resistance character of the amisiCPRNAplum-7 that no plant was infected (Figure 3). The basic approach used to analyze the down-regulation of the PPV genome target is the detection of RNAi accumulated in these plums [18]. In addition to these small RNAi, the DNA methylation of the virus transgene engineered is associated with posttranscriptional mechanisms. In order to better characterize the RNA silencing that occurred, we analyzed the two major components implicated, accumulated RNAi and the virus transgene. Figure 6 shows that these clones accumulate small RNAs and in parallel, there is some evidence that these RNAi are associated with the DNA methylation of the engineered virus transgene. Similar to those results we already observed in plum tree [18], the patterns revealed in Figure 3 and Figure 6 demonstrate that there are 3/10 amisiCPRNA-plums that mediate silencing resistance to PPV. The blockade of the systemic spread of PPV is related to plum defense involving the virus transgene methylated and the RNAi accumulated in these clones.

### 2.4. RNAi Technology for Protecting Perennial Plants

Referring to these studies, PPV spread in *N. benthamiana* has a short life cycle (weeks) when compared to plum trees, known to grow over years. Under four dormancy cycles, PPV, inoculated through grafting, periodically moved up and down from roots to scions. In the same way, mobile RNAi is running through the vascular system. Resistant clones block virus movement because silencing specifically interferes with viral RNA. The RNaseIII enzyme Dicer complex did not allow the development of symptoms [9,13]. To date the appearance of sporadic spots in a few replicates of the amisiCPRNA-plum3 and 4 was unpredicted, however the recovery reaction fit to a late development stage as already indicated in other plant models [26]. Recognized as weakly infected plants (Table 2) the recovery reaction was confirmed through RT/PCR detection of PPV (Figure 3). Rarely observed in greenhouse tests [27], graft-inoculation of HoneySweet plum in field also allowed PPV to cause a few spots in leaves close to grafting point [28,29]. PPV introduced in perennial that moved through the vascular system was interfered with in this way. Regardless of mosaic symptom that systemically developed in susceptible plants, symptomless plants and those with leaves showing sporadic spots reflect the resistance character in perennials accumulating ami- and si-RNA. Over four dormancy cycles, lessons learned, from these perennial plants, significantly showed that these RNAi contributes to the blockade and notably the degradation of the PPV genome (Figure 7).

After transcription, the two respective PriamiCPRNA and PriamisiCPRNA precursors penetrate in the cytoplasm via exportin enzyme. Sliced by AGO-Dicer enzyme into small RNAs, AGO-RISC with the guide strand binds to the viral RNA.

A- Simplified interpretation of RNA mediated silencing pathway in amiCPRNA plum

amiCPRNA-mediated target recognition inhibits the viral genome translation and can cause the viral RNA degradation. Small RNAs contribute to the dsRNA amplification. Amplified RNAi spreads to neighboring cells.

B- Simplified interpretation of RNA mediated silencing pathway in amisiCPRNA plum

amisiCPRNA-mediated target recognition includes amiCPRNA and siRNA. amiCPRNA with the single strand guide binds to AGO-RISC. However, siRNA (a small dsRNA) is loading in AGO-RISC that discards the passenger strand and binds to the viral RNA via the guide strand. The two small RNAs contribute to the inhibition of the viral genome translation and replication. The viral RNA is degraded into small RNAs that contribute to the new dsRNA formation via RDR6 and the tasiRNA induces histone modification. Amplified dsRNA also spreads to neighboring cells to mediate the mobile silencing in whole plant.

## 3. Discussion

To examine genetically engineered resistance in woody perennials, we used as a virus challenge PPV which is the causal agent of an important quarantine disease [1,30]. Research plays an important part in the development of plant breeding programs in response to the stone-fruit industry demands [3,30]. HoneySweet plum is the first woody plant reaching the goal of high level, stable, long-term resistance to PPV [6,7,8]. Besides this clone, there is another version of resistant plum, B14 plant obtained from the PPV CP gene intron-hairpin-RNA construct (ihpRNA) that silenced PPV RNA [18,20]. The efficiency and ability to protect plum trees in field conditions [6,7,8,9,10,11,12,13,14,15,16,28,29] encouraged us to further study this resistance mechanism.

The use of the CP gene can be considered a model viral cistron for producing virus resistant perennial plants [31]. To support or refute our hypothesis about silencing-mediated resistance to PPV infection, we expanded our expertise in an open challenge to engineer amiRNA in plants and verify the potential ability of amiRNA to protect plants [11,12,13,14,15,16,19,21]. The idea to combine ami-siRNA into a gene construct came from the natural contribution of each RNAi molecule for the transcriptional and posttranscriptional control of mRNA in eukaryotes [32,33,34,35,36,37,38,39,40,41].

In this study, we made a genetic modification based on both modes of RNAi silencing, in a natural host of PPV. From the design of RNAi constructs to the applicability of the technology, it is important to consider the involvement of different parameters. Among these were, first, the time-consuming of the assays (four dormancy cycles), secondly, the aggressiveness of PPV, and third, plant physiology linked with temperature, light, humidity, and dormancy. Also, taking in account that PPV is a severe pathogen but it does not generally kill plum-trees [1]. In light of results collected from the first bud-break, there was evidence that PPV was successfully inoculated. Since successful PPV inoculation was clear, preliminary results that pointed out the lesser number of infected plums were significant. Emerging in fact at the early stage of PPV challenging assays, we started to identify resistance or lack thereof through either the lesser number of infected plants or the higher number of uninfected plants.

Consistently, the presence (diseased leaves) or absence of PPV mRNA (resistant clone) gave the first indications of the PPV-RNAi interactions. The molecular machinery including the DICER enzyme complexes to trigger the dsRNA precursor sliced it into siRNA following a clear path for a sustainable resistance. In order to form the RISC, these siRNA were captured by AGO proteins. Beyond that, siRNA guides RISC to target the homologous PPV mRNA sequence that is degraded in fine (Figure 7). We may disregard the amiRNA construct in which the RNAi approach failed. Let us see what happened with the amisiCPRNA plants. In order to understand the difference between the resistance phenotype shared by the experimental plants and the resistance that occurred in the natural hosts, we compare the immune phenotype shown in *N. benthamiana* [19] with the present resistant plum trees graft-inoculated with PPV, that blocked the movement of PPV genome from root to shoot. Referring to the preliminary studies in *N. benthamiana* [19], the efficiency of silencing in herbaceous plants (more than 98% of immune plants) relies in one part on the phloem tissue wherever PPV is spreading and secondly to the short life cycle (weeks). Conversely silencing in plum-trees (3/10 plants), a perennial plant growing over years, could not completely block PPV that was graft inoculated because PPV moves throughout the xylem where RNAi does not accumulate. Serving as a bridge between *N. benthamiana* and *P. domestica*, silencing developed by the resistant plants blocked the systemic spread of PPV. As we show in Figure 6 and Figure 7, resistant clones that accumulated ami- and siRNA blocked the virus movement by having interfered with the viral RNA translation and degradation. As a model, the amisiCPRNA-plum behaved similarly to HoneySweet plum and did not allow to the plant metabolism to express symptoms [42].

The unpredicted appearance of a few spots in recovered clones is not yet been clearly understood. Concerning both the amisiCPRNA-plum-3 and-4, recovery is related to plant physiology (late development stage) [26]. With the long duration of these assays, perennial plants should be adapted to the variable factors in the high containment in greenhouse. Epigenetics is a biological phenomenon encompassing eukaryotic adaptation and recognized as related to the plant genome [43]. Because greenhouse assays required repeatable results, particular care is need for maintaining plants. If these clones were able to resist to PPV under these conditions the genetic modification expressed through the methylation status of their transgene strongly induced the occurrence of RNAi silencing. Less perceptible, the amiCPRNA plums also reacted to PPV infection. They delayed the development of a recovery reaction to the third dormancy cycle. When compared to the phenotype developed by the amisiCPRNA-plum3 and 4, it revealed a poor challenge to PPV infection because the reaction is only visible on the tip, that was confirmed to be virus-free (Figure 5).

Plum trees transformed with amisiCPRNA possess the two major hallmarks of silencing, first the ability to self-amplify ami and siRNA and secondly, to spread via the vascular tissues in the entire plants. The scenario is different with the amiCPRNA plum because since the precursor is transcribed, it is sliced into amiRNA that is captured by the AGO and guided to the viral RNA. In the light of self-amplification, the insert did not have any potentially methylated sites. It means that they could not be similarly amplified like in those amisiCPRNA plants (Figure 7). In the light of silencing, RISC can interact with these amiRNAs that were guided to the targeted PPV mRNA. Insidiously, the ratio between degraded and intact mRNA molecule in amiCPRNA plum-trees was lower because these plants were severely diseased. Under these considerations, PPV RNA is still able to replicate, translate, and move through the entire plants.

Avoiding unsubstantiated speculation, we have sought for years a consistent character of a resistant perennial plum tree. It is interesting to note that during the greenhouse test period, the resistant clone should not show any PPV symptom, neither any positive serological nor molecular detection test; in terms of RNAi detection, the presence of two or multiple band; and the occurrence of the viral transgene methylation [44]. Last, but not least, the tasiRNA from the RNAi pool is among the key-molecules [45] because it is amplified by the RDR6 and guided methylation enzyme to maintain silencing in the entire plants [46]. These criteria shared in any silencing studies led to understanding the robust molecular machinery acting as the source of stability and durability of resistance to virus infection [28,29].

Since the first development of HoneySweet plum [6,7,8], the genetic engineering approach took a great importance in our research. Use of plant or virus gene constructs are expanding [47,48], and ongoing research is producing new PPV resistant transgenic *Prunus* cultivars [49].

## 4. Materials and Methods

### 4.1. Gene Constructs

To consider the involvement of small RNA in tackling virus genome replication, rationale approach based onto RNA folding was initiated [22]. Following to the idea to design ami-RNA, we chose an available web/server (Vienna RNA web servers) that was able to rapidly edit the basic folding of the PPV CP genes. Within the concept of consensual sequence and the quality of the stem-loop structure, we decided to select three viral sequences (Figure 1) from the wide range of quest. As indicated in Figure 1, the selected PPV CP sequence have been respectively assembled with three known miRNA that have been used in gene constructs for virus challenging (see the Appendix A). Arguable motifs related to the folding criteria were chosen for designing the amiRNA construct [20]. Our successful works with either the hairpin- [6,7,8] or intron hairpin- CP construct [18,20] that resist to PPV led us to bring some change in the B14 gene construct [20]. These were the explanations for why the pdk intron of the plant expression vector pHellsgate [50] was removed by restriction digestion and replaced by the trio of amiCPRNAs including amiRNA159 (AthmiRNA159, MI0000189), amiRNA171 (AthmiRNA171, MI0000214) and amiRNA157 (AthmiRNA157, MI0000184) constructs (Appendix A) to give the pHellsgate-amisiCP RNA plasmid (Figure 2B). The second construct designed as pHellsgate-amiCPRNA is homologous to pHellsgate-amisiCPRNA excepted the deletion of both CP sequences flanking the trident amiRNA (Figure 2A). In order to transform plants, the two respective gene constructs were cloned into the *Agrobacteria tumefaciens* vector [24]. Supplemental data to Figure 1 are shown in Appendix A.

### 4.2. Plant Transformation and Selection

To develop the plum model, hypocotyl slices of *Prunus domestica Stanley* cv. were treated with the recombinant *Agrobacteria tumefaciens* [24] harboring either the plasmid vectors pHellsgate-amisiCPRNA or the pHellsgate-amiCPRNA (Figure 2A). Plum clones were selected through their ability to develop on kanamycin-containing media. After rooting, plantlets were transferred into pods and acclimatized in greenhouse (Agreement 2000 of 28/10/2015, Haut Conseil des Biotechnologies, about the use of genetically modified organisms applied in Education, Research and Development).

### 4.3. Molecular Analysis of Plants

To verify the transgene content of each plant, leaves (1gr) of each clone were harvested, then ground into liquid nitrogen for extracting either the total genomic DNA [51] or the total RNA. PCR via the inclusion of a primer pair (5FWDCP and 3REVCP) [52] was used to target the CP gene for the amisiCPRNAplums. For the amiCPRNA, the 5nptII and 3nptII pair was used to amplify the NPTII gene marker plants (not shown) [52].

### 4.4. Challenging Assays and Symptomatology

Plums transformed with the two plasmid vectors designed as pHellsgate-amiCPRNA or the pHellsgate-amisiCPRNA which accumulated either artificial miRNA (amiCPRNA) or ami-and siCPRNA, respectively, were studied through their potential ability to challenge sharka disease [1,2]. A protocol consisting of graft transgenic buds in the susceptible *Prunus marianna* rootstocks was set up for propagating clones into a high containment greenhouse [6,18] (Agreement of 31/01/2018, Agriculture Ministry, about the use of quarantine pests applied in Education, Research and Development). Developed plantlets within eight to twelve leaf stages were then graft-inoculated with PPV strain M [6,18]. In disregard to symptom appearance on leaves one month after inoculation, all plantlets were transferred into a cold room. Following the three months of dormancy in cold, all plants were reset in the greenhouse. Six weeks after bud-breaking, infected plants started to show symptoms, particular attention was paid to the rootstock section that should be diseased. At this stage, the susceptibility of some plants was distinguished by others that could remain symptomless.

### 4.5. Serological and Molecular Analyses

Expecting the successful passage of PPV in scions, leaves from either rootstock or scions were sampled from six weeks following to bud-breaking. They were ground and plant sap were analyzed through DAS-ELISA according to [53]. Virus protein levels of were detected following the manufacturer procedure (LCA, La Rochelle, France). OD values were evaluated by readings via a BioTek Epoch plate reader (Winooski, Vermont, USA) at 405 nm. Histograms that marked the statistical difference in infected plants allowed distinguishing susceptible from resistant clones (not shown). In disregard to the level of PPV mRNA in plum-trees, PPV genome was screened from backgrounds by reverse transcription-polymerase chain reaction (OneStep RT/PCR kit, Qiagen, Valencia, CA, USA) [27,54], using the couple of primers (80Nib and 660REV) reported in Table 1. An aliquot of the total RNA (1 µg) was used as template and incubated in 25 µL of reagent consisting to 1X OneStep RT/PCR buffer, 50 µM dNTPs, 1 U of mixed enzyme (reverse transcriptase and Taq polymerase) and 1 µL of 10 pm of each primer. Due to the expected size of the amplicon (880 bp), the RT was modified at 50 °C during 40 min, then the RNA/cDNA was denatured at 95 °C for 15 min. PCR was performed with 40 cycles of denaturation at 95 °C for 30 s, annealing 52 °C for 30 s and an amplification at 72 °C for 1 min. A final extension step was carried out at 72 °C for 10 min. The amplified fragment of 880 bp spanning COOH of PPV-Nib (nuclear inclusion b) and the medium part of CP was verified onto agarose gel electrophoresis (1%) with Tris-borate-EDTA pH 8.3.

### 4.6. Down Regulation Studies

Down-regulation of PPV genome replication by RNAi silencing can be perceived through two biological phenomena related to the virus DNA-methylated transgene [18,27,45] and the detection of RNAi in plant tissue analyzed [27,55]. Inhibition of PPV RNA replication linked with the blockade of virus genes expression is correlated with the dominant phenotypes exhibited by the clones studied.

### 4.7. DNA Methylation

The aim of the methylation study was to look at how much of the virus transgene was naturally mutated by the plant methylases. The modification of the transgene status was based on the comparative analyses within two isoschizomers *BfuCI* and *MboI* recognizing the same restriction sites GATC. 2 µg of DNA was over-digested with either the isoschizomer *MboI* or the *BFCuI* restriction enzymes overnight. The methylated status of the engineered PPVCP cistron clones was validated by PCR. An aliquot of the over-digested DNA was amplified with the use of a couple of primers (340FWD and 660REV) flanking the two GATC sites potentially methylated [27]. Expectedly, there is no amplicon produced with the restriction fragments resulting from *MboI*, because all DNA was cut. However, PCR analysis of those cut with *BfuCI* has the key-role to precisely indicate the methylated status of the PPVCP gene introduced in the plum genome [18,27]. An amplicon of 425 bp was consequently detected by PCR (Figure 6).

### 4.8. RNAi Detection

As we demonstrated in our previous studies, RNAi detection followed the routine procedure of total RNA extraction. Fresh leaves (500 mg) of the transformed *Prunus domestica* clones were collected and ground with mortar and pestle with nitrogen liquid. The powder was then treated according to the RNA kit (Norgen Biotek Corp, Thorold, ON, Canada), as recommended by the manufacturer. After elution from column, RNA extraction was improved through the additive step of phenol/chloroform treatment that facilitated the suspension of the pelleted RNA. For the detection of RNAi, an aliquot of the total RNA was fractionated on denaturing 16% urea-polyacrylamide gel Tris-borate-EDTA (TBE). Electro-transferred to Hybond-NX membrane (GE Healthcare), the amisiCPRNA was hybridized with α-32P- dCTP (Perkin Elmer, Waltham, MA, USA) radiolabeled CP probe (Figure 2B, Figure 6A). The extracted amiCPRNA was probed with miRNA157, miRNA159 and miRNA171 labeled with a γ- 32P- ATP (Perkin Elmer, Waltham, MA, USA) radiolabeled probes (Figure 2C). Hybridization was revealed through autoradiography (GE Healthcare MP) [18,27,55].

## 5. Conclusions

We conclude that these results demonstrate additional RNAi approaches developed with ami and siRNA engineered in plum-trees that efficiently block the systemic spread of PPV in the natural *Prunus domestica* host.

## Figures and Tables

**Figure 1 plants-08-00565-f001:**
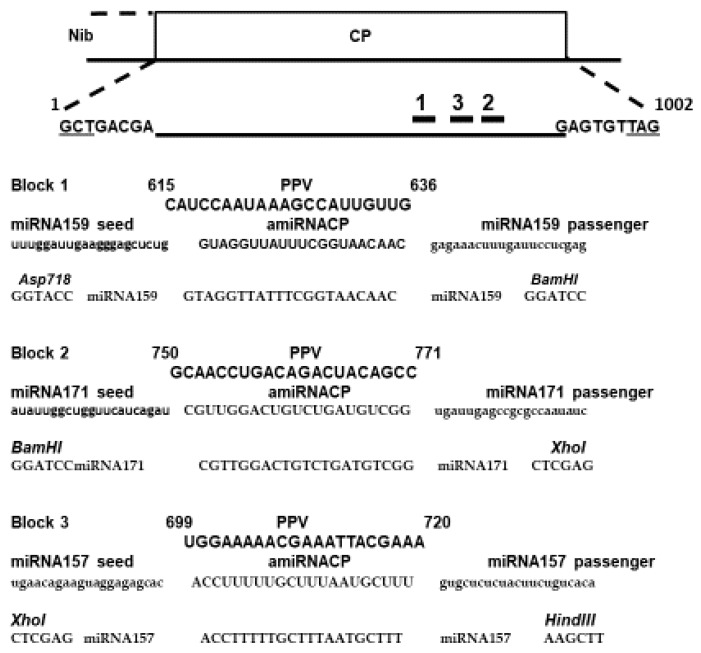
Target PPV CP genome (X16415.1) and the designed amiCPRNA constructs. The GCT and TAG underlined trinucleotides represent the amino acid residue Glycine (NH2) and the stop codon of the CP.

**Figure 2 plants-08-00565-f002:**
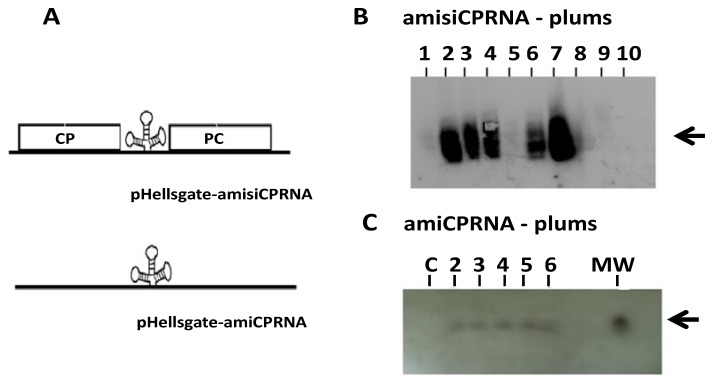
(**A**) Scheme of the gene construct harbored by the recombinant pHellsgate- amisiCPRNA and pHellsgate-amiCPRNA. (**B**) Autoradiograms showing results of Northern-blot done with total RNA extracted from young leaves of the 10 clones (lanes 1 to 10) obtained from plum transformation with pHellsgate-amisiCPRNA. The membrane was hybridized with α-32P-dCTP PPV CP amplicon probe (Table 1). (**C**): Results of Northern blot analysis with total RNA extracted from young leaves of 6 plums including 5 clones obtained from plum transformation with pHellsgate-amiCPRNA, the conventional plum BO70146 used as control (**C**) and a set of synthetic single-stranded RNA oligonucleotides 17, 21, and 25 residues long (MW) as the molecular weight markers (New England Biolabs, Ipswich, MA, USA). After separation onto 16% PAGE. RNA was transferred onto membrane and hybridized with a mixture of probes including miRNA 157, 159, 171, and a miRNA molecular weight marker probe labeled with γ- 32P-ATP (Table 1). The numbers (upper lanes) represent the clones studied. Arrow (right margin) indicates the expected bands detected.

**Figure 3 plants-08-00565-f003:**
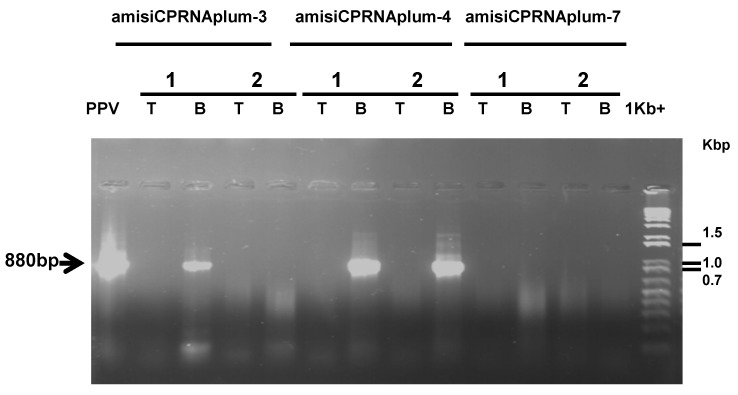
Agarose gel electrophoretic analysis of the OneStep RT/PCR of the total RNA extracted from leaves of plants challenged with *Plum pox virus* (PPV). The amplicon of 880 bp (arrow in the left margin), using the couple of primers (80Nib and 660REV) reported in Table 1, represents the expected band detected from the infected conventional plum BO70146 (lane PPV). Lanes 1 and 2 represent two different plants of the challenged clones (amisiCPRNAplum-3, -4, and -7). The samples T or B mean that leaves were sampled either from tip (T) or the bottom part (B) of the scion. Lane 1Kb+ DNA ladder (Invitrogen, Thermo Fisher Scientific, Waltham, MA, USA) represents the molecular weight markers in Kbp.

**Figure 4 plants-08-00565-f004:**
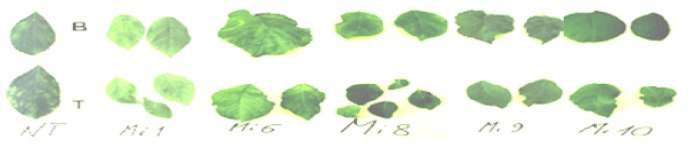
Leaf samples from the two levels B and T of the amiCPRNAplum clones pointed out in Table 3 and those of the conventional BO70146 (NT) for comparison.

**Figure 5 plants-08-00565-f005:**
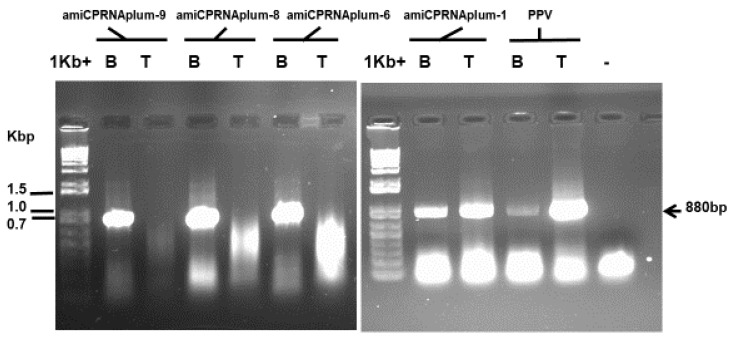
Agarose gel electrophoretic analysis showing the systemic movement of PPV (lane PPV, used as control). Each lane represents the results of OneStep RT/PCR analysis of a total RNA extracted from leaves (shown in Figure 4) sampled either from tip (T) or the bottom part (B) of the individual clones amiCPRNAplum-1, -4, -6, and -9 (cited in the upper lanes). Lane (-) as mock (RNA was substituted by water in the reagent mixture) and lane PPV were used for comparison. The arrow (right margin) indicates the predicted amplicon of 880 bp from the infected conventional BO70146 (lanes PPV). Lane 1Kb+ DNA ladder (Invitrogen, Thermo Fisher Scientific, Waltham, MA, USA) represents the molecular weight markers in Kbp.

**Figure 6 plants-08-00565-f006:**
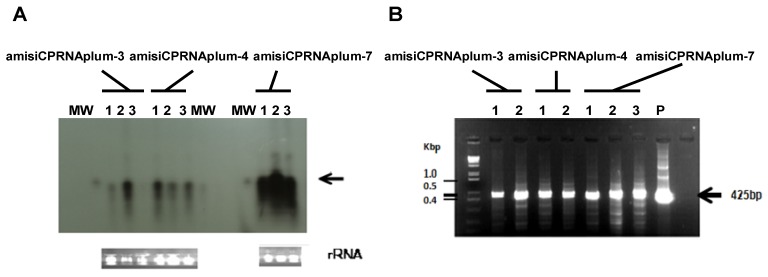
(**A**) Autoradiogram showing the accumulated RNAi detected in leaves of 3 different plants (Lanes 1 to 3) of each clone (amisiCPRNAplum-3, -4, and -7) challenged with PPV. 20 µg of total RNA was fractionated onto PAGE. Transferred onto membrane, RNAi was probed with a PPVCP probe labeled with α- 32P-dCTP. Lane MW represents the miRNA molecular weight marker (indicated by the arrow in the right margin) hybridized with a γ-32P-ATP miRNA probe. Ethidium bromide staining of rRNA was used as loading control (bottom panels). (**B**): Agarose gel showing the DNA methylation analysis of the virus CP transgene introduced in amisiCPRNAplums. Genomic DNA was extracted from leaves of 2 different plants (Lanes 1 and 2) of each clone (amisiCPRNAplum-3, -4) and 3 plants of the amisiCPRNAplum-7 challenged with PPV. After over-digestion of DNA with either the isoschizomer *MboI* (not shown) or the *BFCuI* restriction enzymes, the methylated status of the PPVCP cistron engineered in these clones was verified by PCR. With the use of a couple of primers (340FWD and 660REV, Table 1) flanking the two GATC sites potentially methylated, an amplicon of 425 bp (arrow in the right margin) indicates the expected band amplified, as got from the uncut DNA control (lane P, uncut pHellsgate-amisiCPRNA plasmid). Lane 1Kb DNA ladder (Invitrogen, Thermo Fisher Scientific, Waltham, MA, USA) represents the molecular weight markers in Kbp.

**Figure 7 plants-08-00565-f007:**
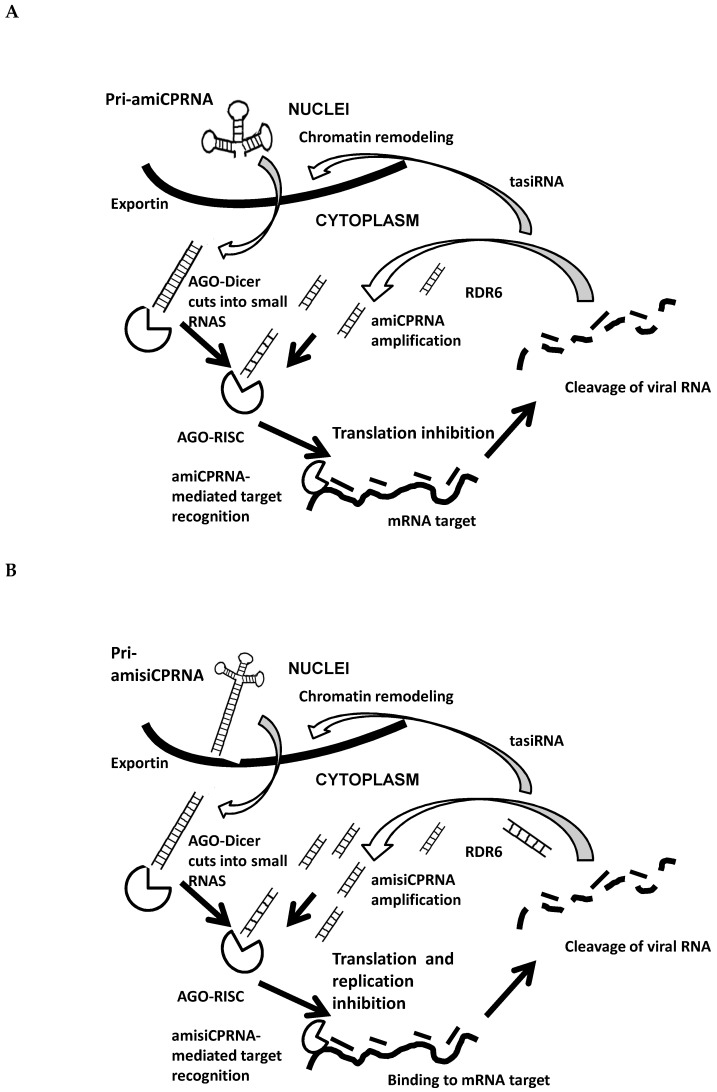
Comparison of the RNAi mediated silencing pathway in amiCPRNAplums (**A**) and amisiCPRNAplums (**B**).

**Table 1 plants-08-00565-t001:** List of primers utilized in these studies.

Primers	Sequences	Cistron Sources Experiments
5FWDCP	AAGCTGAYGAAAGRGAGGACGAG	PPV CP detection 32P Probe
3REVCP	CTACACTCCCCTCACACCGAGGAA	PPV CP detection 32P Probe
80Nib	TTGGGTTCTTGAACAAGCACC	Nib PPV detection
340FWD	CAACTCAAACGCGCTAGTCAAC	CP Methylation
660REV	ATACGCTTCAGCCACGTTACTG	CP Methylation PPV detection
miRNA157	GTGCTCTCCTACTTCTGT	amiRNA detection 32P probe
miRNA159	TAGAGCTTCCCTTCAATCCT	amiRNA detection 32P probe
miRNA171	ATCTGATGAACCTGCCAAT	amiRNA detection 32P probe
miRNA marker	AAATCTCAACCAGCCACTGCT	Molecular weight marker Probe

**Table 2 plants-08-00565-t002:** Results of challenging assays (number of plants infected/plants inoculated) of the different clones infected with PPV strain M after the 1st and the 4th dormancy cycles.

**amiCPRNA-PLUMS**							
**CLONES**	**2**	**6**	**8**	**9**	**11**	**12**	**15**
1^st^ CYCLE	4/10	5/10	3/4	3/4	4/10	3/4	4/6
4^TH^ CYCLE	4/4	5/5	3/3	3/3	4/4	3/3	4/4
**amisiCPRNA-PLUMS**							
**CLONES**	**2**	**3**	**4**	**6**	**7**	**10**	
1^st^ CYCLE	4/4	1/4	2/3	6/6	0/4	4/7	
4^TH^ CYCLE	4/4	1/4	1/3	6/6	0/4	4/4	

**Table 3 plants-08-00565-t003:** OD values obtained from leaves collected from the tip section of a few amiCPRNA plums that apparently developed a recovery reaction following to the 4th dormancy cycle.

Clones	Bottom	Tip
amiCPRNA-plum1	+++	1.23
amiCPRNA-plum6	+++	0.55
amiCPRNA-plum8	+++	0.23
amiCPRNA-plum9	+++	0.0
amiCPRNA-plum10	+++	0.23

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
