# Peer review of "Innovative RNAi Strategies and Tactics to Tackle Plum Pox Virus (PPV) Genome in Prunus domestica-Plum"

_plants, 2019, doi:10.3390/plants8120565_

Round 1

Reviewer 1 Report

The research work is interesting and designed in an appropriate way, but corrections are necessary to improve the manuscript in the Material and Methods, and Results sections, overall in terms of: fluidity of speech; cross-references between the text and the tables / figures, order of the last ones. In the Results section, several parts could be moved into discussions’. Otherwise, Results and Discussion could be united under a single section. A check of the cross-references between the text and the final list is necessary. Some observations are given in the attached pdf file in the form of comments.

Author Response

RESPONSES TO REVIEWER 1

The research work is interesting and designed in an appropriate way, but corrections are necessary to improve the manuscript in the Material and Methods, and Results sections, overall in terms of: fluidity of speech; cross-references between the text and the tables / figures, order of the last ones. I

We are grateful to your appreciations about the works done. As suggested, we have reviewed the manuscript and rearranged it. To comply with the fluidity of results claimed by the reviewer, we modified the order of Figures and Tables that support the results shown. For instance, Fig.2 was divided in three section, 2A: in order to characterize the systemic spread of PPV in plum-trees, a comparative molecular detection of PPV was shown in the 3 clones amisiCPRNAplum, qualified as resistant clones. As recommended, we have modified many phrases including the confused links with Table 1. In order to help the reader to follow up what happened with these plants, Fig.2B,  2C and Table 3  related with amiCPRNAplum were modified.

the Results section, several parts could be moved into discussions’. Otherwise, Results and Discussion could be united under a single section.

According to the recommendations we did not combined the results and discussion section because they can be separately clarified, We agree that we were dispersed in some part of the results section to give some remarks or comments that could serve as matter for discussion. So consequently, we did move some item like that in the discussion. Presently we agree that the significant results are released in the right place.

A check of the cross-references between the text and the final list is necessary. Some observations are given in the attached pdf file in the form of comments.

We apologized about the cross-references between text and the cited literature because we were fallen in wrong list when we didn’t really take care about the mismatch caused by the automatic conversion of the document in pdf format (for instance [51, 52] in the former version). We apologized about such uncontrolled saving.

Reviewer 2 Report

Summary of research:

The manuscript describes Innovative RNAi strategies and tactics to tackle plum 2 pox virus (PPV) genome in Prunus domestica-plum. The authors performed computational target prediction to identify two gene construct from CP cistron, and designed amiCPRNA(artificial miRNA) and amisiCPRNA (artificial mi and si-RNA). These gene construct were then tested to find the potential effects on RNAi production and PPV infection. Overall, I think that this is an interesting study, however, I have comments that would strengthen the manuscript. I would like the authors to address these, which are described below.

1. The author should add a concluding sentence, as it will help the audience better understand the entire story of the paper.
2. Line 32-33: Sentence is not clear. Authors should make it more clear.
3. Figure 1Ba and 1Bb. Please label the figure for northern blot as per the figure legend.
4. Figure 2: Lable the band size of the agarose gel image. Also indicate the marker used in the method and result section.
5. As a general comment for all the images, and respective figure legend: All should be labeled indicating band size and sample details.
6. Please modify figure 4B, either show the black and white or colored gel image

Author Response

Responses to reviewer 2:

We  thank you for the suggestions and corrections that improve the manuscript.

Enclosed are the corrections about: Page 1, l.32-33: Introduction (section)

…. “Throughout the European Union researchers at diverse institution are seeking solutions to control the disease incidence but the divergence of economic concerns between country is still challenging ”...   Was changed to 

Throughout the European Union countries, researchers are seeking solutions to control the disease incidence but the economic imbalance between each and other state led to the lack of harmonization.

Conclusion:

We conclude that these results demonstrate additional RNAi approaches developed with ami and siRNA engineered in plum-trees that efficiently block the systemic spread of PPV in the natural Prunus domestica host.

Even if we have already mentioned the molecular weight markers either the 1KB+ DNA molecular weight marker for Fig.2 or the 1KB DNA molecular weight marker for Fig.4 in the caption.

For this time, we have included all details recommended as like for the molecular weight marker in the Materials and Methods section.

As mentioned, we indicated the band size of the expected amplicon in Fig.2 and 4. All sample details were added in the incomplete legends.

In Fig.4B, the pattern of the 1KB molecular weight marker DNA (represented in different bands) is fainter so we have resolved the image to be clearer (that was why it was grey/white). For this time, as quested, we improved the Fig.4B, according to the better resolution to show the image in black/white.

Reviewer 3 Report

This MS reports the use of 2 RNAi strategies to silence the CP gene of Plum Pox virus (PPV) in plum, previously used successfully in Nicotiana by the same authors. As a whole, the technical approach is appropriate for the purpose of the MS and results increases the tools to tackle PPV in a non-model species. Specially, it should be taking into account the difficulties to work with trees. However, in my opinion more efforts should be done in order to explain clearly the experiments conducted and the obtained results. For this purpose, the complete MS should be carefully rewritten. In this sense, I have some comments that might be considered by the authors:

- The structure of the MS should be prepared according to the “Instructions for Authors”, as: 1- Intoduction, 2- Results, 3- Discussion, 4-Materials and Methods, and 5-Conclusions (optional).

- Seems that the references are not well numbered. For instance, number 29 (Varsha et al. 2001) seems that should be number 22 (Petri et al., 2011).

- The manuscript must be revised by a native English speaker.

- Introduction [lines 28-78]

The ideas are not properly explained. Specially, the previous work in Nicotiana (Ravelonandro et al., 2013 Acta Hort). It should be detailed here the results observed in this work, as the starting point of this one.

- Materials and Methods [lines 79-147]

This section should be moved to the 4th position.

According to the instructions, “M&M should be described with sufficient detail to allow others to replicate and build on published results”. A detailed description or citation of previous methods should be included. For instance, lines 81-98 should be rewritten to explain properly the constructs employed. If I am not wrong, the constructs employed here to transform plum are the ones previously employed to transform Nicotiana. So, it should be just cited. The sentence “Preliminary experiments… used as model [28]” (lines 92-93) should be removed. The complete paragraph should be rewritten just explaining briefly with citations the constructs and the method of plum transformation. Morever the name of the cultivar employed should be included.

The results obtained should be removed from this section. For instance, in the lines 98-96 the number of clones selected.

Lines 99-104: The details of the PCR protocol should be included: primer sequences, program..

Lines 106-117: The sentence from line 106-109 could be deleted as it was already explained before. I do not understand clearly the time employed for the phenotyping process. How often you subject the plants to an artificial winter? Just after each phenotyping cycle?

Line 126: The table with the primer sequences should be cited here.

Lines 127-131 should be deleted as they are redundant.

Lines 132-137: A more detailed explanation of the digestion and PCR should be included. The primer sequences should be included in the table.

Line 138-147: The first sentence should be deleted. The probe sequences should also be included in the table.

- Results: [lines 148-388]

Subsections 3.1 and 3.2 are describing Results, but 3.3 and 3.4 are more related with Discussion. In general, the complete section should be carefully revised, specially ambiguity in explanations and excessive elucubration should be avoided. For instance, in lines 150-151 authors explain some kind of method, not described before, that should be properly described in M&M section. Similarly, some studies are explained in lines 292-296, but no citation is present.

Author Response

Comments and Suggestions for Authors

This MS reports the use of 2 RNAi strategies to silence the CP gene of Plum Pox virus (PPV) in plum, previously used successfully in Nicotiana by the same authors. As a whole, the technical approach is appropriate for the purpose of the MS and results increases the tools to tackle PPV in a non-model species. Specially, it should be taking into account the difficulties to work with trees. However, in my opinion more efforts should be done in order to explain clearly the experiments conducted and the obtained results. For this purpose, the complete MS should be carefully rewritten. In this sense, I have some comments that might be considered by the authors:

We approve the comments and critics of the reviewer 2 and we want to point out that these results are significantly scientific about RNAi mechanisms in perennial hosts. To fight against a severe virus qualified as a quarantine pest, we do not have yet available solutions to help stone fruit industry. Control measures remain a hot spot for sharka disease. In addition to the previous findings with HoneySweet plum, we show here plum-trees accumulating either ami-, or ami –siRNA to tackle PPV. Academic resources were obtained with the experimental Nicotiana benthamiana. Besides scientific resources, based onto the obtention of these herbaceous plant models, important tools and protocols were developed for assessing PPV resistance.

According to the quest, we rewrote the manuscript and highlighted these findings in the introduction section. To transfer the technology to Prunus domestica was the challenge. The virus resistance study in perennials required a good knowledge about tree physiology. As we previously indicated upper, PPV resistance in greenhouse required official agreement (we added the reference in the manuscript), the use of high number of plant replicates ensures the repeatability of the assays. While we mentioned that plum-trees infected to PPV do not die. The alternate repetition of plant growth and cold in an artificial way can reduce the viability of perennials.  In addition, it requires more space either for plant development or in cold room for dormancy cycles, because the objective is to identify the clone phenotype that should be uniformly expressed. As a lasting experience, these assays were also dependent to the environmental parameters including light, humidity and heat.  The general idea is to get homogenate behavior however we know that PPV in perennials is a concern because the virus distribution is irregular. The is why we did many repeatable analyses (serological and molecular detection) that eliminate any confusion. Pioneering the study with the known HoneySweet plum resistant to PPV, we have in hands either plant control (susceptible and resistant) or tools and methodology to characterize plants. The detection of PPV in perennials follows a standard protocol to record PPV infection. Cold dormancy is a natural requirement for perennials because bud-breaking is the phenomenon that allows PPV to move from root to shoots. That is why we need to do artificial cold because the virus resistance is dependent of the blockade of the systemic spread of PPV. So the manuscript structure is based on the procedure used. Greenhouse testing methods allow to obtain the resistant plum clones. RNAi mediated silencing via ami-, or ami –siRNA to tackle PPV genome was the objective of the manuscript. There are two key-points that represent the differences between resistant N.benthamiana and plum—trees.

First the resistance phenotype in N.benthamiana is immunity.  Excepted with aphid-inoculation, we cannot demonstrate any immune character in plum-trees. We recognize that the major difficulty to infect plum-trees is related with the irregular distribution of PPV.  According to the protocol we developed via graft inoculation, the high resistance character was shown by the amisiCPRNAplum-7 that is perceived through a high number of dormancy cycle to be able to block the systemic spread of PPV.

 Second the statistical data, immune N.benthamiana  is around 98% however plum trees have a lower score 1/10 highly resistant and 2/10 recovered. There is no any rational perception about such results because the efficiency of RNAi differed from plant to plant. In regard to the technology, the role of amiRNA when compared to ami and siRNA reveals that it is not enough to tackle PPV genome however the combination of both was verified in the amisiCPplum-7 clone. The manuscript was written in that sense. This is the major objective of this manuscript. This is the novelty. We understood why both reviewers insisted about the development of the Material and Methods and the clarification of the findings so we improved the manuscript as recommended for disseminating helpful information. 

The structure of the MS should be prepared according to the “Instructions for Authors”, as: 1- Intoduction, 2- Results, 3- Discussion, 4-Materials and Methods, and 5-Conclusions (optional).

As we already communicated to reviewer 1, we have reviewed the manuscript and rearranged as quested. We improved through introduction, results, discussion and materials and methods.

We avoided to comments the results section.

Thank you for the suggestion, however we did not give any conclusion because the results can serve as a baseline to promote research on PPV control. Even if these results can provide an added value of plant biotechnology!

Seems that the references are not well numbered. For instance, number 29 (Varsha et al. 2001) seems that should be number 22 (Petri et al., 2011).

As we communicated above, we apologized about the false cross-references between text and the cited literature because, in addition, we were fallen in wrong list when we saved the document in the pdf format. We did take care for this time.

The manuscript must be revised by a native English speaker.

We  have improved the manuscript via the correction by a native English.

  Introduction [lines 28-78]

The ideas are not properly explained. Specially, the previous work in Nicotiana (Ravelonandro et al., 2013 Acta Hort). It should be detailed here the results observed in this work, as the starting point of this one.

We have rewritten the manuscript and clarified through your suggestion the results developed in N.benthamiana in the introduction section.  This manuscript related to PPV resistance gave immune plants. The successful use of ami-siRNA versus amiRNA led us to pursue the ongoing work at that time (2013) based onto the obtention of plums transformed with the same gene constructs.  The rationale was to use innovative tools including amiRNA to fight aginst PPV. Because HoneySweet plum is accumulating siRNA. We expanded the idea to develop perennial plants within ami, amisi. The challenge was the inclusion of amiRNA and the technology transfer in plum-tree. To tackle the quarantine PPV that represents an economical treat for stone-fruit industry, we develop plant biotechnology approach that is an alternate approach to classical breeding. We are acknowledging the expertise of reviewers who recommended how and why the results presented in this manuscript should be improved.

- Materials and Methods [lines 79-147]

This section should be moved to the 4th position.

As recommended, we moved Materials and Methods in the 4th position

According to the instructions, “M&M should be described with sufficient detail to allow others to replicate and build on published results”. A detailed description or citation of previous methods should be included. For instance, lines 81-98 should be rewritten to explain properly the constructs employed.

We brought more accuracy in the Material and Methods section that contains applicable data for laboratory and field research consisting to directly control and inspect PPV replication in perennials. As we cited above, the underlying technologies were already published as we clearly cited in references. We also rewrote some key-points like the methylation study that strongly support silencing mechanisms. Fig. 5 provides some details about the difference with ami and amisiRNA effects.

Considered as a new approach in perennials, the development of new varieties with new traits were developed in this paper.

Thinking that the details about the amiRNA constructs were unnecessary, we provided the details in this new version. And we added in the supplemental data, the cloning of these constructs in the pHellsgate plant transformation vectors.

We hope that we comply with the quest, to educate and to inform scientists about protocols, gene engineering…

We are feeling that any end-users (breeders, pathologists, growers) can collect some ideas and protocols for improving their knowledge.

If I am not wrong, the constructs employed here to transform plum are the ones previously employed to transform Nicotiana. So, it should be just cited. The sentence “Preliminary experiments… used as model [28]” (lines 92-93) should be removed. The complete paragraph should be rewritten just explaining briefly with citations the constructs and the method of plum transformation. Morever the name of the cultivar employed should be included.

Yes, this manuscript is about plums transformed with the same constructs, and accumulating amiRNA and both small RNAs (amiRNA and siRNA).  In the past we just provided a scheme but following your quest, we provided the details. A supplemental data was added about the cloning in pHellsgate vector. We thought that these details are enough.

The results obtained should be removed from this section. For instance, in the lines 98-96 the number of clones selected.

We did it as quested

Lines 99-104: The details of the PCR protocol should be included: primer sequences, program..

We provided it in the Materials and Methods section and added more sequences in Table 2

Lines 106-117: The sentence from line 106-109 could be deleted as it was already explained before. I do not understand clearly the time employed for the phenotyping process. How often you subject the plants to an artificial winter? Just after each phenotyping cycle?

Yes, we deleted it.

Having rewritten the paper, we also clarified above why cold is determinant to record PPV infection in perennials. This manuscript related specifically the virus resistance using amiRNA and both am-siRNA in perennials. The phenotypes are expressed in another way than the binary reaction as “immune” or “susceptible” in herbaceous plants.

Line 126: The table with the primer sequences should be cited here.

We did it

Lines 127-131 should be deleted as they are redundant.

They were deleted.

Lines 132-137: A more detailed explanation of the digestion and PCR should be included. The primer sequences should be included in the table.

In order to understand the results in Fig 4A, we added a scheme (b)  to follow up the methylation status. The use of these primers was already published [ 27 ], as quested, we inserted their sequences in Table 2 (List of primers)

Line 138-147: The first sentence should be deleted. The probe sequences should also be included in the table.

 We did both, deletion of lines 138-147 and listed sequences of probes.

- Results: [lines 148-388]

Subsections 3.1 and 3.2 are describing Results, but 3.3 and 3.4 are more related with Discussion. In general, the complete section should be carefully revised, specially ambiguity in explanations and excessive elucubration should be avoided. For instance, in lines 150-151 authors explain some kind of method, not described before, that should be properly described in M&M section. Similarly, some studies are explained in lines 292-296, but no citation is present.

We rewrote

Round 2

Reviewer 3 Report

MS plants-632734

First of all, I would like to recognize the author´s effort in reviewing the MS. In general, results are shown in a clearer order and quality of writing has been also improved leading to a much more comprehensible paper. However, in my opinion there are some important details that should be taking into account. I would like to make it clear that I think it is a very interesting work and that it requires a great effort. For this reason, in my opinion, it would not be fair that it could not be published due to defects in writing.

I have also some additional comments about this new version of the MS:

Figures should be improved for clarity purposes. I am not sure if the figures having different parts will be joined during the editing process after the approval, but in my opinion they are not ready for publication yet. Figure 1 has 3 parts, but they appear as different figures with different legends. For instance, Fig 1B and Fig 1C has also a and b parts. So, it is a bit confusing. Maybe all of them could be included as just 1 figure in 1 page, with different parts but all together. Figure legends are in general too large, making a bit difficult the reading process. The exception is the Fig4B legend, that should be rewritten to express the real meaning of the figure, as “Study about the genomic DNA” is too ambiguous. The section explaining the “silencing studies” is still a bit confusing. Authors analysed the pattern of methylation of the PPV CP gene and the detection of RNAi in plant tissue to detect the down-regulation of PPV genome replication by RNAi silencing (as explained in M&M section (lines 479-484). However, in the Results section (lines 257-289) the explanation is not clear. Lines 277-289 are written as part of a M&M section, but the explanation of the results observed in the figures is not clear. It would be greatly appreciated if authors could rewrite this subsection for clarity purposes. Also the sentence in line 490 is not complete.

Minor points:

Line 49: “were” should be replaced by “was” Line 64: A dot should be added after [1,2]. Lines 69-71: The sentence “and to assess their effects on PPV infection. The thinking behind these transgenic plums was to assess the effects of these RNAi to PPV infection in perennial plants.” should be replaced by “and to assess their effects on PPV infection in perennial plants.” Line 75: A comma should be added after amiRNA Line 86: The reference to Fig 1Ba should be removed here, because the figure is not related with the sentence. Line 161: A dot should be added after “copies)” Line 258: The sentence “Tables 1 and 3, and Fig. 2 show the results.” should be deleted and they should be just cited in the proper place. Figure 4Bb is not needed and should be avoided, as it is already part of the 1Ba.

Author Response

RESPONSES TO REVIEWER 3

We would like to thank you for critical reading and comments concerning the manuscript. We really appreciate the valuable and useful corrections. We have improved our paper according to the recommendations about figures and items which were confusing. We have considered and integrated all corrections regarded as minor points. Enclosed are explanations and changes about the different points you raised.

Figures

Figure 1 is solely about the target PPV CP sequences used to design the amiCPRNA constructs. Due to the size of this figure, we avoided to include 4 panels in this figure.

Figure 2 has 3 parts. Panel 1 represents the scheme of the two recombinant pHellsgate-amiCPRNA and pHellsgateamisiCPRNA. Panel 2 and 3 are about the small RNAs detection in amisiCPRNA- and amiCPRNA-plums

Figure 3 is unchanged because there is a readable difficulty linked to the different resolution when we attempt to join in one the 3 panels about agarose gel electrophoresis and sampled leaves of amiCPRNAplums.

Figure 4 was improved through the junction of the 2 panels about RNAi detection and methylation studies in amisiCPRNAplums.

Figure 5 is unchanged.

Changes in the text

Page 2, l.82: “After cloning of the amiCPRNA constructs” was changed to “ After ensuring design for the amiCPRNA constructs (Fig.1), these were cloned into…”

Page 2, l.86: “In further analysis…(Fig.2A)” was changed to “In further analysis of RNAi accumulated in each clone, a variable amount of RNAi was detected in amisiCPRNA-plums (Fig.2A)”

Page 9, l.255-270: We have rewritten the item “silencing studies” and changed to

“2.3. Down-regulation of the PPV genome replication by RNAi silencing

Serological detection of PPV allowed to point out that a few copies of these clones amisiCPRNAplum-3 and -4 were infected with PPV (Table 1 and Fig.3A). The repetition of these analyses 15 days later allowed to confirm the recovery reaction developed by the amisiCPRNAplum-3, -4 and the high resistance character of the amisiCPRNA-plum7 that no any plant was infected (Fig 3A). The basic approach used to analyse the down-regulation of the PPV genome target is the detection of RNAi accumulated in these plums [18]. In addition to these small RNAi, the DNA methylation of the virus transgene engineered is associated with posttranscriptional mechanisms. In order to better characterize the RNA silencing occurred, we analysed the two major components implicated, accumulated RNAi and the virus transgene. Fig.4 shows that these clones accumulate small RNAs and in parallel, there is some evidence that these RNAi are associated with the DNA methylation of the virus transgene engineered. Similar to those results we already observed in plum-tree [18], the patterns revealed in Fig. 3A and 4 demonstrate that there are 3/10 amisiCPRNA-plums that mediate silencing resistance to PPV. In fine, the blockade of the systemic spread of PPV is related to plum defense involving the virus transgene methylated and the RNAi accumulated in these clones.”

Page 9-10, l.273-288: We have rewritten the caption and changed to

“Figure 4. Panel A: Autoradiogram showing the accumulated RNAi detected in leaves of 3 different plants (Lanes 1 to 3) of each clone (amisiCPRNAplum-3, -4 and -7) challenged with PPV. 20 µg of total RNA was fractionated onto PAGE. Transferred onto membrane, RNAi was probed with a PPVCP probe labeled with α- 32P-dCTP. Lane MW represents the miRNA molecular weight marker (indicated by the arrow in the right margin) hybridized with a γ-32P-ATP miRNA probe. Ethidium bromide staining of rRNA was used as loading control (bottom panels). Panel B: Agarose gel showing the DNA methylation analysis of the virus CP transgene introduced in amisiCPRNAplums. Genomic DNA was extracted from leaves of 2 different plants (Lanes 1 and 2) of each clone (amisiCPRNAplum-3, -4) and 3 plants of the amisiCPRNAplum-7 challenged with PPV. After over-digestion of DNA with either the isoschizomer MboI (not shown) or the BFCuI restriction enzymes, the methylated status of the PPVCP cistron engineered in these clones was verified by PCR. With the use of a couple of primers (340FWD and 660REV) flanking the two GATC sites potentially methylated, an amplicon of 425bp (arrow in the right margin) indicates the expected band amplified, as got from the uncut DNA control (lane P, uncut pHellsgate-amisiCPRNA plasmid). Lane 1Kb DNA ladder (Invitrogen, Thermo Fisher Scientific, Waltham, MA, USA) represents the molecular weight markers in Kbp.”

Page 15, l.489-499: We have rewritten the Materials and Methods section about DNA methylation and changed to

“DNA methylation

The aim of the methylation study was to look at how much of the virus transgene was naturally mutated by the plant methylases. The modification of the transgene status was based on the comparative analyses within two isoschizomers BfuCI and MboI recognizing the same restriction sites GATC. 2 µg of DNA was over-digested with either the isoschizomer MboI or the BFCuI restriction enzymes overnight. The methylated status of the engineered PPVCP cistron clones was validated by PCR. An aliquot of the over-digested DNA was amplified with the use of a couple of primers (340FWD and 660REV) flanking the two GATC sites potentially methylated [27]. Expectedly, there is no amplicon produced with the restriction fragments resulting from MboI, because all DNA was cut. However, PCR analysis of those cut with BfuCI has the key-role to precisely indicate the methylated status of the PPVCP gene introduced in the plum genome [18,27]. An amplicon of 425bp was consequently detected by PCR.”

As indicated upper, we have improved the paper, we hope that the revised manuscript comply with the recommendations you suggested. Again, we are thanking you about the valuable comments and suggestions.
